# Propensity Score Alignment of Unpaired Multimodal Data

**Johnny Xi**[*]
Department of Statistics
University of British Columbia
Vancouver, Canada
`johnny.xi@stat.ubc.ca`

**Jana Osea**
Valence Labs
Montreal, Canada
`jana@valencelabs.com`

**Zuheng (David) Xu** [*]
Department of Statistics
University of British Columbia
Vancouver, Canada
`zuheng.xu@stat.ubc.ca`

**Jason Hartford**
Valence Labs
London, UK
`jason@valencelabs.com`

## Abstract

Multimodal representation learning techniques typically require paired samples to learn shared representations, but collecting paired samples can be challenging in fields like biology, where measurement devices often destroy the samples. This paper presents an approach to address the challenge of aligning unpaired samples across disparate modalities in multimodal representation learning. We draw an analogy between potential outcomes in causal inference and potential views in multimodal observations, allowing us to leverage Rubin's framework to estimate a common space for matching samples. Our approach assumes experimentally perturbed samples by treatments, and uses this to estimate a propensity score from each modality. We show that the propensity score encapsulates all shared information between a latent state and treatment, and can be used to define a distance between samples. We experiment with two alignment techniques that leverage this distance—shared nearest neighbours (SNN) and optimal transport (OT) matching—and find that OT matching results in significant improvements over state-of-the-art alignment approaches in on synthetic multi-modal tasks, in real-world data from NeurIPS Multimodal Single-Cell Integration Challenge, and on a single cell microscopy to expression prediction task.

## 1 Introduction

Large-scale multimodal representation learning techniques such as CLIP [Radford et al., 2021] have lead to remarkable improvements in zero-shot classification performance and have enabled the recent success in conditional generative models. However, the effectiveness of multimodal methods hinges on the availability of *paired* samples—such as images and their associated captions—across data modalities. This reliance on paired samples is most obvious in the InfoNCE loss [Gutmann and Hyvärinen, 2010, van den Oord et al., 2018] used in CLIP [Radford et al., 2021] which explicitly learns representations to maximize the true matching between images and their captions.

While paired image captioning data is abundant on the internet, paired multimodal data is often challenging to collect in scientific experiments. For instance, unpaired data are the norm in biology for technical reasons: RNA sequencing, protein expression assays, and the collection of microscopy

---

[*]Work done during an internship at Valence Labs

38th Conference on Neural Information Processing Systems (NeurIPS 2024).

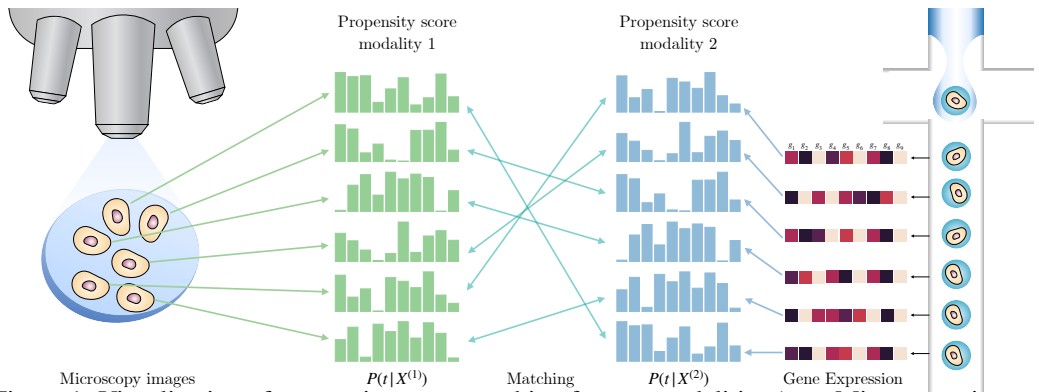

Figure 1: Visualization of propensity score matching for two modalities (e.g., Microscopy images and RNA expression data). We first train classifiers to estimate the propensity score for samples from each modalities; the propensity score reveals the shared information $p(t|z_i)$, which allows us to re-pair the observed disconnected modalities. The matching procedure is then performed within each perturbation class based on the similarity bewteen the propensity scores.

images for cell painting assays are all destructive processes. As such, we cannot collect multiple different measurements from the same cell, and can only explicitly group cells by their experimental condition. If we could accurately match unpaired samples across modalities, we could use the aligned samples as proxies for paired samples and apply existing multimodal learning techniques.

In this paper, we formalize this setting by viewing each modality as a *potential* measurement, $X^{(1)}(Z) \in \mathcal{X}^{(1)}, X^{(2)}(Z) \in \mathcal{X}^{(2)}$, of the same underlying latent state $Z \in \mathcal{Z}$, where we are only able to make a single measurement for each sample unit (e.g. an individual cell). The task is to reconcile (*match*) unpaired observations $x^{(1)}$ and $x^{(2)}$ with the same (or maximally similar) $z$. Estimating the latent, $Z$, is hopelessly underspecified without making unverifiable assumptions on the system, and furthermore, $Z$ may still be sparse and high-dimensional, leading to inefficient matching. This motivates the need for approaches that use only the observable data.

We identify two major challenges for this problem. First, measurements are often made in very different spaces $\mathcal{X}^{(1)}$ and $\mathcal{X}^{(2)}$ (e.g., pixel space and gene expression counts), which make defining a notion of similarity across modalities challenging. Second, the measurement process inevitably introduces modality-specific variation that can be impossible to disentangle from the relevant information ($Z$). For example in cell imaging, we would not want the matching to depend on irrelevant appearance features such as the orientation of the cell or the lighting of the plate.

In this paper, we address these challenges by appealing to classical ideas from causal inference [Rubin, 1974], in the case where we additionally observe some label $t$ for each unit, e.g., indexing an experiment. By making the assumption that $t$ perturbs the observations via their shared latent state, we identify an observable link between modalities with the same underlying $z$. Under conditions which we discuss in Section 2, the propensity score, defined as $p(t|Z)$, is a transformation of the latent $Z$ that satisfies three remarkable properties (Proposition 3.1): (1) it provides a common space for matching, (2) it is fully identifiable via classification on individual modalities, and (3) it maximally reduces the dimension of $Z$, retaining only the information revealed by the perturbations.

The practical implementation of the methodology (as illustrated in Fig. 1) is then straightforward: we train two separate classifiers, one for each modality, to predict the treatment $t$ applied to $X^{(i)}$. We then match across modalities based on the similarity between the predicted probabilities (the propensity score) within each treatment group. This matching procedure is highly versatile and can be applied to match labeled observations between any modalities for which a classifier can be efficiently trained. However, since the same sample unit does not appear in both modalities, we cannot use naive bipartite matching. To address this, we use soft matching techniques to estimate the missing modality for each sample unit by allowing matching to multiple observations. We experiment with two recent matching approaches: shared nearest neighbours (SNN) matching [Lance et al., 2022, Cao and Gao, 2022] and optimal transport (OT) matching Villani [2009].

In our experiments, we find that OT matching with distances defined on the proposenity score leads to significant improvement on matching and a downstream cross-modality prediction task on both synthetic and real-world biological data. Notably, our prediction method, which leverages the soft matching to optimize an OT projected loss, outperforms supervised learning on the true pairs on CITE-seq data from the NeurIPS Multimodal Single-Cell Integration Challenge [Lance et al., 2022]. Finally, we applied our method to match single-cell expression data (from a PeturbSeq assay [Dixit et al., 2016]) with single cell crops of image data [Fay et al., 2023]. We find improved generalization in predicting the distribution of gene expression from the cell imaging data in with unseen perturbations.

## 1.1 Related Work

**Unpaired and Multimodal Data** Learning from unpaired data has long been considered for image translation [Liu et al., 2017, Zhu et al., 2017, Almahairi et al., 2018], and more recently for biological modality translation [Amodio and Krishnaswamy, 2018, Yang et al., 2021]. In particular, Yang et al. [2021] also takes the perspective of a shared latent variable for biological modalities. This setting has been studied more generally for multi-view representation learning [Gresele et al., 2020, Sturma et al., 2023] for its identifiability benefits.

**Perturbations and Heterogeneity** Many methods in biology treat observation-level heterogeneity as a nuisance dimension to globally integrate, even when cluster labels are observed [Butler et al., 2018, Korsunsky et al., 2019, Foster et al., 2022]. This is sensible when clusters correspond to noise rather than the signal of interest. However, it is well known in causal representation learning that heterogeneity—particularly heterogeneity arising from perturbations—has theoretical benefits in constraining the solution set [Khemakhem et al., 2020, Squires et al., 2023, Ahuja et al., 2023, Buchholz et al., 2023, von Kügelgen et al., 2023]. There, the benefits (weakly) increase with the number of perturbations, which is also true of our setting (Proposition 3.2). In the context of unpaired data, only Yang et al. [2021] explicitly leverage this heterogeneity in their method, while Ryu et al. [2024] treat it as a constraint in solving OT. Specifically, Yang et al. [2021] require their VAE representations to classify experimental labels in addition to reconstructing modalities, while our method is simpler, only requiring the classification objective. Notably, Yang et al. [2021] treat our objective as a regularizer, but our theory suggests that it is actually primarily responsible for the matching performance. Our experiment results coincide with the theoretical insights; requiring reconstruction, as in a VAE, led to worse matching performance with identical model architectures.

**Optimal Transport Matching** OT is a common tool in single-cell biology. In cell trajectory inference, the unpaired samples are gene expression values measured at different time points in a shared (metric) space. OT matching minimizes this shared metric between time points [Schiebinger et al., 2019, Tong et al., 2020]. Recent work [Demetci et al., 2022] extends this to our setting where each modality is observed in separate metric spaces by using the Gromov-Wasserstein distance, which computes the difference between the metric evaluated within pairs of points from each modality [Demetci et al., 2022]. In concurrent work, this approach was recently extended to ensure matching within experimental labels [Ryu et al., 2024]. In addition to these "pure" OT approaches, Gossi et al. [2023] use OT on contrastive learning representations, though this approach requires matched pairs for training, while Cao et al. [2022] use OT in the latent space of a multi-modal VAE.

## 2 Setting

We consider the setting where there exist two potential views, $X^{(e)} \in \mathcal{X}^{(e)}$ from two different modalities indexed by $e \in \{1, 2\}$, and experiment $t$ that perturbs a shared latent state of these observations. This defines a jointly distributed random variable $(X^{(1)}, X^{(2)}, e, t)$, from which we observe only a single modality, its index, and label, $\{x_i^{(e_i)}, e_i, t_i\}_{i=1}^n$.[2] We aim to match or estimate the samples from the missing modality, which corresponds to the realization of the missing random variable. Since $t$ is observed, in practice we match observations *within* the same label class $t$.

Formally, we assume each modality is generated by a common latent random variable $Z$ as follows:

$$t \sim P_T, \ Z^{(t)} \mid t \sim P_Z^{(t)}, \ U^{(e)} \sim P_U^{(e)}, \ U^{(e)} \perp\!\!\!\perp Z, \ U^{(e)} \perp\!\!\!\perp U^{(e')}, \ X^{(e)} \mid t = f^{(e)}(Z^{(t)}, U^{(e)}), \ (1)$$

---

[2]We will denote random variables by upper-case letters, and samples by their corresponding lower-case letter.

where $t$ indexes the experimental perturbations, and we take $t = 0$ to represent a base environment. $U^{(e)}$ represents the modality-specific measurement noise that is unperturbed by $t$, and also independent across samples. The structural equations $f^e$ are deterministic after accounting for the randomness in $Z$ and $U$: it represents the measurement process that captures the latent state. For example, in a microscopy image, this would be the microscope and camera that maps a cell to pixels.

**Comparison to Multimodal Generative Models**   Our setting is technically that of a multimodal generative model with latent perturbations. However, by focusing on matching rather than generation, we are able to make significantly weaker and more meaningful assumptions while still ensuring the theoretical validity of our method. Without the effects of the perturbation, our Eq. (1) is essentially the same as [Yang et al., 2021, Equation 1] in an abstract sense. However, in order to fit the generative model, it is required to formulate explicit models over $f^{(e)}$ and $P_Z^{(t)}$, which requires specifying the function class (e.g., continuous) and the space of $Z$ (e.g., $\mathbb{R}^d$) as assumptions, even in universal approximation settings. In contrast, since we will not directly fit the model Eq. (1), we do not make any technical assumptions about the generative model. Instead, we will make the following assumptions on the underlying data generating process itself.

**Key Assumptions**   Our theory makes the following assumptions about the data generating process.

(A1) $t \not\perp\!\!\!\perp Z$, and $t \perp\!\!\!\perp U^{(e)}$. In words, $t$ has a non-trivial effect on $Z$, but does not affect $U^{(e)}$, implying that interventions target the common underlying process without affecting modality-specific properties. For example, an intervention that affects the underlying cell state, but not the measurement noise of the individual modalities.

(A2) Injectivity of $f^{(e)}$: $f^{(e)}(z, u) = f^{(e)}(z', u') \implies (z, u) = (z', u')$. In words, each modality captures enough information to distinguish changes in the underlying state.[3]

(A1) ensures that the conditional distribution $t \mid X^{(e)}$ is identical for $e = 1, 2$. (A2) then ensures that $t \mid X^{(1)} \overset{\mathrm{d}}{=} t \mid X^{(2)} \overset{\mathrm{d}}{=} t \mid Z$, which allows us to estimate the conditional distribution $t \mid Z$ with observed data alone. Though sharp assumptions are required for the theory, versions replaced with approximate distributional equalities intuitively also allow for effective matchings when combined with our soft matching procedures in practice. A particular relaxation of (A1) when combined with OT matching is described in Appendix A.

## 3   Multimodal Propensity Scores

Under (1), if $Z$ were observable, an optimal matching can be constructed by simply matching the samples with the most similar $z_i$. However, the prerequisite of inverting the model and disentangling $Z$ is arguably more difficult than the matching problem itself. In particular, $Z$ is unidentifiable without strong assumptions on Eq. (1) [Xi and Bloem-Reddy, 2023], and even formulating the identifiability problem requires well-specification of the model as a prerequisite. We take an alternative approach that is robust to these problems, by using the perturbations $t$ as an observable link to reveal information about $Z$. Specifically, we show that the propensity score

$$\pi(z) := P(t|Z = z) \in [0, 1]^{T+1}, \tag{2}$$

is identifiable as a proxy for the latent $Z$ under our assumptions of the data generating process. This is a consequence of the injectivity of $f^{(e)}$, since it will be that $\pi(Z) = \pi(X^{(e)})$, $e = 1, 2$, indicating that we can compute it from either modality. Not only does the propensity score reveal shared information, classical causal inference theory [Rubin, 1974] states that it captures *all* information about $Z$ that is contained in $t$, and does so minimally, in terms of having minimum dimension and entropy. Since $t$ contains the only observable information that is useful for matching, the propensity score is hence an optimal compression of the observed information. We collect these observations into the following proposition.

**Proposition 3.1.** *In the model described by Eq.* (1)*, further assume that $f^{(e)}$ are injective for $e = 1, 2$. Then, the propensity scores in either modality is equal to the propensity score given by $Z$,*

---

[3]Note that the injectivity is in the sense of $f$ as a function of both $u$ and $z$, which allows observations that have a shared $z$ but differ by their value in $u$, and the function remains injective. For example, rotated images with the exact same content can have a shared $z$, but remain injective due to the rotation being captured in $u$.

*i.e.,* $\pi(X^{(1)}) = \pi(X^{(2)}) = \pi(Z)$ *as random variables. This implies*

$$I(t, Z \mid \pi(Z)) = I(t, Z \mid \pi(X^{(e)})) = 0, \tag{3}$$

*for each $e = 1, 2$, where $I$ is the mutual information. Furthermore, any other function $b(Z)$ satisfying $I(t, Z \mid b(Z)) = 0$ is such that $\pi(Z) = f(b(Z))$.*

The proof can be found in Appendix C. Practically, Proposition 3.1 shows that computing the propensity score on either modality is equivalent to computing it on the unobserved shared latent, which means that it is identifiable, and thus estimable, from the observations alone. Furthermore, the estimation does not require modified objectives or architectures for joint multimodal processing, instead they are simple and separate classification problems for each modality. Finally, $t$ does not affect $U^{(e)}$ by assumption, and thus the propensity score, being a representation of the information in $t$, discards the modality-specific information that may be counterproductive to matching. Therefore, even if $Z$ were observed, it may be sensible to match on its propensity score instead.

**Number of Perturbations** Note that point-wise equality of the propensity score $\pi(z_1) = \pi(z_2)$ does not necessarily imply equality of the latents $z_1 = z_2$, due to potential non-injectivity of $\pi$. For example, consider $t \in \{0, 1\}$, then $\pi(z)$ is a compression to a single dimension $z \to p(t = 1 \mid z)$. Intuitively, collecting data from more perturbations improves the amount of information contained in the label $t$. If the latent space is $\mathbb{R}^d$, the propensity score necessarily compresses information about $Z^{(t)}$ if the latent dimension exceeds the number of perturbations, echoing impossibility results from the causal representation learning literature [Squires et al., 2023].

**Proposition 3.2.** *Let $Z^{(t)} \in \mathbb{R}^d$. Suppose that $P_Z^{(t)}$ has a smooth density $p(z|t)$ for each $t = 0, \ldots, T$. Then, if $T < d$, the propensity score $\pi$, restricted to its strictly positive part, is non-injective.*

The proof can be found in Appendix C. Note the above only states an impossibility result when $T < d$. More generally, it can be seen from the proof of Proposition 3.2 that the injectivity of the propensity score depends on the injectivity of the following expression in $z$:

$$g(z) = \begin{bmatrix} \log(p(z|t=1)) - \log(p(z|t=0)) \\ \vdots \\ \log(p(z|t=T)) - \log(p(z|t=0)) \end{bmatrix}, \tag{4}$$

which then depends on the latent process itself. If the above mapping is non-injective, this represents a fundamental indeterminacy that cannot be resolved without making strong assumptions on point-wise latent variable recovery. As we have already established in Proposition 3.1, the propensity score contains the maximal shared information across modalities. Nonetheless, collecting data form a larger number of perturbations is clearly beneficial for matching, since $g$ in Eq. (4) is injective if any of the subset of its entries are.

## 4 Estimation and Matching

For the remainder of the paper, we drop the notation $e$ and use $(x_i, t_i)$ to denote observations from modality 1, and $(x_j, t_j)$ to denote observations from modality 2. Given a multimodal dataset with observations $\{(x_i, t_i)\}_{i=1}^{n_1}$ and $\{(x_j, t_j)\}_{j=1}^{n_2}$, we wish to compute a matching matrix (or coupling) between the two modalities. We define a $n_1 \times n_2$ matching matrix $M$ where $M_{ij}$ represents the likelihood of $x_i$ being matched to $x_j$. Since $t$ is observed, we always perform matching only within observations with the same value of $t$, so that in practice we obtain a matrix $M_t$ for each $t$.

Our method approximates the propensity scores by training separate classifiers that predicts $t$ given $x$ for each modality. We denote the estimated propensity score by $\pi_i$ and $\pi_j$ respectively, where

$$\pi_i \approx \pi(x_i) = P(T = t \mid X_i^{(e)} = x_i). \tag{5}$$

This yields the transformed datasets $\{\pi_i\}_{i=1}^{n_1}$ and $\{\pi_j\}_{j=1}^{n_2}$, where $\pi_i$, $\pi_j$ are in the $T$ dimensional simplex. We use this correspondence to compute a cross-modality distance function:

$$d(x_i, x_j) := d'(\pi_i, \pi_j). \tag{6}$$

In practice, we typically compute the Euclidean distance in $\mathbb{R}^T$ of the logit-transformed classification scores, but any metric over a bijective transformation of the propensity scores are also theoretically

valid. Given this distance function, we use existing matching techniques to constructing a matching matrix. In our experiments, we found that OT matching gave the best performance, but we also evaluated Shared Nearest Neighbour matching; details of the latter can be found in Appendix B.

**Optimal Transport Matching**   The propensity score distance allows us to easily compute a cost function associated with transporting mass between modalities, $c(x_i, x_j) = d'(\pi_i, \pi_j)$. Let $p_1, p_2$ denote the uniform distribution over $\{\pi_i\}_{i=1}^{n_1}$ and $\{\pi_j\}_{j=1}^{n_2}$ respectively. Discrete OT aims to solve the problem of optimally redistributing mass from $p_1$ to $p_2$ in terms of incurring the lowest cost. Let $C_{ij} = c(x_i, x_j)$ denote the $n_1 \times n_2$ cost matrix. The Kantorovich formulation of optimal transport aims to solve the following constrained optimization problem:

$$\min_M \sum_i^{n_1} \sum_j^{n_2} C_{ij} M_{ij}, \quad M_{ij} \geq 0, \quad M\mathbf{1} = p_1, \quad M^\top \mathbf{1} = p_2. \tag{7}$$

This is a linear program, and for $n_1 = n_2$, it can be shown that the optimal solution is a bipartite matching between $\{\pi_i\}_{i=1}^{n_1}$ and $\{\pi_j\}_{j=1}^{n_2}$. We refer to this as exact OT; in practice we add an entropic regularization term, resulting in a soft matching, that ensures smoothness and uniqueness, and can be solved efficiently using Sinkhorn's algorithm. Entropic OT takes the following form:

$$\min_M \sum_i^{n_1} \sum_j^{n_2} C_{ij} M_{ij} - \lambda H(M), \quad M_{ij} \geq 0, \quad M\mathbf{1} = p_1, \quad M^\top \mathbf{1} = p_2, \tag{8}$$

where $H(M) = -\sum_{i,j} M_{ij} \log(M_{ij})$, the entropy of the joint distribution implied by $M$. This approach regularizes towards a higher entropy solution, which has been shown to have statistical benefits [Genevay et al., 2018], but nonetheless for small enough $\lambda$ serves as a computationally appealing approximation to exact OT.

## 5   Downstream Tasks

The matching matrix $M$ can be seen as defining an empirical joint distribution over the samples in each modality. The OT approach in particular makes this explicit. Each row is proportional to the probability that each sample $i$ from modality (1) is matched to sample $j$ in modality (2), i.e., $M_{i,j} = P(x_j|x_i)$. We can thus use $M$ to obtain pseudosamples for any learning task that uses paired samples by $(x_i, \hat{x}_j)$, where $\hat{x}_j$ is obtained by sampling from the conditional distribution defined by $M$, or by a suitable conditional expectation, e.g., the barycentric projection (conditional mean) as $E_M[X_j \mid X_i = x_i] = \sum_j M_{i,j} x_j$. In what follows, we describe a cross-modality prediction method based on both barycentric projection and stochastic gradients according to $M_{i,j}$.

**Cross-modality prediction**   We can use the matching matrix to design a method for cross-modality prediction/translation. The following MSE loss corresponds to constructing a prediction function $f_\theta$ such that the barycentric projection $E_M[f_\theta(X_j) \mid X_i = x_i]$, under $M$ minimizes the squared error for predicting $x_i$:

$$\mathcal{L}(\theta) := \sum_i (x_i - \sum_j M_{i,j} f_\theta(x_j))^2. \tag{9}$$

However, this requires evaluating $f_\theta$ for all $n_2$ examples from modality (2) for each of the $n_1$ examples in modality (1). In practice, we can avoid this cost with stochastic gradient descent by sampling from modality (2) via $M_{i.}$ for each training example (1). To obtain an unbiased estimate of $\nabla_\theta \mathcal{L}$, we need two independent samples from modality (2) for each sample from modality (1),

$$\nabla \mathcal{L}(\theta) \approx -2\left(x_i - f_\theta(\dot{x}_j)\right) \nabla_\theta f_\theta(\ddot{x}_j) \quad \dot{x}_j, \ddot{x}_j \sim P(x_j|x_i). \tag{10}$$

By taking two samples as in Eq. (10), we get an unbiased estimator of $\nabla \mathcal{L}(\theta)$, whereas a single sample would have resulted in optimizing an upper-bound on equation (9); for details, see Hartford et al. [2017] where a similar issue arises in the gradient of their causal effect estimator. We thus refer to prediction models trained via Eq. (10) as *unbiased*.

# 6 Experiments

We present a comprehensive evaluation of our proposed methodology on three distinct datasets: (1) synthetic paired images, (2) single-cell CITE-seq dataset (simultaneous measurement of single-cell RNA-seq and surface protein measurements) [Stoeckius et al., 2017], and (3) Perturb-seq and single-cell image data. In the first two cases, there is a ground-truth matching that we use for evaluation, but samples are randomly permuted during training. This allows us to exactly compute the quality of the matching in comparison to the ground truth. The final dataset is a more realistic setting where ground truth paired samples do not exist, and matching becomes necessary in practice. In this case, we compute distributional metrics to compare our proposed methodology against other baselines.

**Experimental Details** All models for the experiments are implemented using `Torch v2.2.2` [Paszke et al., 2017] and `Pytorch Lightning v2.2.4` [Falcon and PyTorch Lightning Team, 2023]. The classifier used to estimate the propensity score is always a linear head on top of an encoder $E_i$, which is specific to each modality and dataset. All models are saved at the optimal validation loss to perform subsequent matching. Shared nearest neighbours (SNN) is implemented using `scikit-learn v1.4.0` [Pedregosa et al., 2011] using a single neighbour, and OT is implemented using the Sinkhorn algorithm as implemented in the `pot v0.9.3` package [Flamary et al., 2021]. Both SNN and OT use the Euclidean distance as the metric. Whenever random variation can affect the results of the experiments, we report quantiles corresponding to variation from different random seeds. Additional experimental details are provided in Appendix D.

**Description of Baselines** Our main baseline, which we evaluate against on all three datasets, is matching using representations learned by the multimodal VAE of Yang et al. [2021], which is the only published method that is able to leverage perturbation labels for unpaired multimodal data (they refer to the labels as "prior information"). The standard multimodal VAE loss is a reconstruction loss based on encoder and decoders $E_i$, $D_i$ for each modality, plus a latent invariance loss that aims to align the modalities in the latent space. In our setting, the multimodal VAE loss further includes an additional label classification loss from the latent space of each modality, i.e., encouraging the encoder to simultaneously learn $P(t \mid E_i(x_i))$. This additional objective, which acts as a regularizer for the multimodal VAE, is exactly the loss for our proposed method. To ensure a fair comparison, we always use the same architecture in the encoders $E_i$ of multimodal VAE and in our propensity score classifier. The performance differences between propensity score matching and multimodal VAE then represent the effects of the VAE reconstruction objective and latent invariance objectives. For additional baselines, we also compare against a random matching, where the samples are matched with equal weight within each perturbation as a sanity check. For datasets (1) and (2), we also compare against Gromov-Wasserstein OT (SCOT) [Demetci et al., 2022] computed separately within each perturbation. SCOT uses OT directly by computing a cost function derived based on pairwise distances within each modality, thus learning a local description of the geometry which can be compared between modalities. For the CITE-seq dataset, we also compare against matching using a graph-linked VAE, scGLUE [Cao and Gao, 2022], where the graph is constructed from linking genes with the associated proteins.

**Evaluation Metrics** We use the known ground truth matching to compute performance metrics on datasets (1) and (2). The trace and FOSCTTM [Liu et al., 2019] measure how much weight $M$ places on the true pairing. However, this is not necessarily indicative of downstream performance as similar, but not exact matches are penalized equally to wildly incorrect matches. For this reason, we also measure the latent MSE for dataset (1) and the performance of a CITE-seq gene–to–protein predictive model based on the learned matching for dataset (2). For more details, see Appendix D.1.

## 6.1 Experiment 1: Synthetic Interventional Images

**Data** We followed the data generating process Eq. (1) with a latent variable $Z$ encoding the coordinates of two objects. Perturbations represent different do-interventions on the different dimensons of $Z$. The difference between modalities corresponds to whether the objects are circular or square, and a fixed transformation of $Z$, while the modality-specific noise $U$ controls background distortions.

**Model and Evaluation** We used a convolutional neural network adapted from Yang et al. [2021] as the encoder. We report two evaluation metrics: (1) the trace metric, and (2) the MSE between

|  | Synthetic Image Data | | CITE-seq Data | |
|---|---|---|---|---|
| **Method** | **MSE ($\downarrow$)** **Med (Q1, Q3)** | **Trace ($\uparrow$)** **Med (Q1, Q3)** $\times 10^{-3}$ | **FOSCTTM ($\downarrow$)** **Med (Q1, Q3)** | **Trace ($\uparrow$)** **Med (Q1, Q3)** |
| PS+OT | **0.0316** **(0.0300, 0.0330)** | **18.329** **(17.068, 18.987)** | **0.3049** **(0.3008, 0.3078)** | **0.1163** **(0.1093, 0.1250)** |
| VAE+OT | 0.0324 (0.0316, 0.0350) | 7.733 (7.473, 7.794) | 0.3953 (0.3912, 0.4045) | 0.0814 (0.0777, 0.8895) |
| PS+SNN | 0.0552 (0.0530, 0.0558) | 7.924 (7.569, 9.504) | 0.3126 (0.3121, 0.3160) | 0.0941 (0.0880, 0.0989) |
| VAE+SNN | 0.0622 (0.0571, 0.0676) | 3.116 (2.818, 3.213) | 0.3816 (0.3760, 0.3822) | 0.0612 (0.0588, 0.0634) |
| SCOT | 0.0354 | 0.5964 | 0.4596 | 0.0200 |
| GLUE+SNN | - | - | 0.4412 | 0.0362 |
| GLUE+OT | - | - | 0.5309 | 0.0323 |
| Random | 0.0709 (0.0707, 0.0714) | - | - | - |

Table 1: Alignment metrics results using synthetic interventional image dataset and CITE-seq data.

the matched and the true latents. The latent MSE metric does not penalize close neighbours of the true match (i.e. examples for which $\|z_i - z_i^*\|$ is small) as heavily as the trace metric. These "near matches" will typically still be useful on downstream multimodal tasks.

**Results** In Table 1, metrics are computed on a held out test set over 12 groups corresponding to interventions on the latent position, with approximately 1700 observations per group. A random matching, with weight $1/n$, will hence have a trace metric of of $1/1700 \approx 0.588 \times 10^{-3}$. This implies, for example, that the median performance of PS+OT is approximately 31 times that of random matching. On both metrics, we found that propensity scores matched with OT (PS + OT) consistently outperformed other matching methods on both metrics.

## 6.2 Experiment 2: CITE-Seq Data

**Data** We used the CITE-seq dataset from the NeurIPS 2021 Multimodal single-cell data integration competition [Lance et al., 2022], consisting of paired RNA-seq and surface level protein measurements over 45 cell types. In the absence of perturbations, we used the cell type as the observed label to classify and match within. Note the cell types are determined by consensus by pooling annotations from marker genes/proteins. In most cells, the annotations from each modality agreed, suggesting that the label is independent from the modality-specific noise. We used the first 200 principal components as the gene expression modality, and normalized (but otherwise raw) protein measurements as input.

**Model and Evaluation** We used fully-connected MLPs as encoders. To assess matching, we report (1) the trace, and (2) the Fraction Of Samples Closer Than the True Match (FOSCTTM) ([Demetci et al., 2022], [Liu et al., 2019]) (lower is better, 0.5 corresponds to random guessing). To evaluate against a downstream task, we also compared the performance of random and VAE matching procedures, as well as directly using the ground truth ($M_{ii} = 1$), on predicting protein levels from gene expression. We trained a 2-layer MLP (the same architecture for all matchings) with both MSE loss and the unbiased procedure as described in Section 5 using pseudosamples sampled according to the matching matrix. We evaluated the predictive models against ground truth pairs by computing the prediction $R^2$ (higher is better) on a held-out, unpermuted, test set.

**Results** In Table 1, metrics are computed on a held-out test set averaged over 45 cell types with varying observation counts per group. While interpreting the average trace can be challenging due to group size variations, OT matching on PS consistently outperformed other methods both within and across groups. In these experiments, OT matching on PS was consistently the top performer, often followed by SNN matching on PS or OT matching on VAE embeddings.

We present downstream task performance in Table 2. Note that $R^2$ is computed using the sample average across possibly multiple cell types, which explains why random matching within each cell type results in non-zero $R^2$ (see Appendix D.1). We found that PS + OT matching outperforms other methods on this task. Surprisingly, the PS + OT prediction model performed even better on average than training with the standard MSE loss on ground truth pairings (though confidence intervals

| Method | CITE-seq Data | | PerturbSeq/Single Cell Image Data | |
|---|---|---|---|---|
| | **MSE Loss** $R^2$ **Med (Q1, Q3)** (↑) | **Unbiased Loss** $R^2$ **Med (Q1, Q3)** (↑) | **In Distribution** **KL Med (Q1, Q3)** (↓) | **Out of Distribution** **KL** (↓) |
| Random | 0.138 (0.137, 0.140) | 0.173 (0.170, 0.173) | 58.806 (58.771, 60.531) | 51.310 |
| VAE+OT | 0.149 (0.118, 0.172) | 0.114 (0.079, 0.159) | 55.483 (55.410, 56.994) | 47.910 |
| **PS+OT** | **0.217** **(0.206, 0.223)** | **0.233** **(0.207, 0.250)** | **50.967** **(50.898, 52.457)** | **43.554** |
| True Pairs | 0.224 (0.223, 0.226) | - | - | - |

Table 2: Cross-modal prediction results using CITE-seq data and PerturbSeq/single cell image data including an out of distribution distance evaluation for PerturbSeq/single cell images.

overlap). This highlights the potential benefit of soft (OT) matching as a regularizer, beyond that of simply reconciling most likely pairs: the soft matching effectively averages over modality specific variation from samples with similar latent states in a manner analogous to data augmentation (with an unknown group action).

### 6.3 Experiment 3: PerturbSeq and Single Cell Images

**Data** We collected PerturbSeq data (200 genes) and single-cell images of HUVEC cells with 24 gene perturbations and a control perturbation, resulting in 25 total labels across both modalities. As preprocessing, we embed the raw PerturbSeq counts into a 128-dimensional space using scVI [Lopez et al., 2018] and the cell images into a 1024-dimensional space using a pre-trained Masked Autoencoder [He et al., 2022, Kraus et al., 2023] to train our gene expression and image classifiers.

**Model and Evaluation** We used a fully connected 2-layer MLP as the encoder for both PerturbSeq and cell image classifiers. Similarly to the CITE-seq dataset, we evaluated the matchings based on downstream prediction of gene expression from (embeddings of) images. We used the unbiased procedure to minimize the projected loss Eq. (9) and evaluated on two held-out sets, one consisting of in-distribution samples from the 25 perturbations the classifier was trained on, and an out-of-distribution set consisting of an extra perturbation not seen in training. In the absence of ground truth matching, we assessed three distributional metrics between the actual and predicted gene expression values within each perturbation: the L2 norm of the difference in means, the Kullback-Leibler (KL) divergence, and 1-Wasserstein distance (lower indicates better alignment). We report inverse cell-count weighted averages over each perturbation group. Each metric measures a slightly different aspect of fit—the L2 norm reports a first-order deviation, while the KL divergence is an empirical estimate of the deviation of the underlying predicted distribution, while the 1-Wasserstein distance measures deviations in terms of the empirical samples themselves.

Note that matching is performed using classifiers trained on scVI embeddings, but the cross-modal predictions are generated in the original log transformed gene expression space (i.e. we predicted actual observations, not embeddings). We also evaluated distance measures on an out-of-distribution gene perturbation that was not used in either the matching or training of the translation model.

**Results** We present KL divergence values for in-distribution and out-of-distribution in Table 2.[4] Additional metrics show similar patterns and can be found in Appendix D.4. OT + PS matching consistently outperforms its VAE counterpart both on in-distribution and out-of-distribution metrics, supporting our findings on the CITE-seq data to the case where ground truth pairs are not available.

### 6.4 Validation Monitor

As in our Perturb-seq and cell imaging example, the ground truth matching is typically unknown in real problems. It is hence desirable to have an observable proxy of the matching performance

---

[4]We computation of in-distribution metrics using random subsamples from the test set. The out-of-distribution metric was computed on a small dataset with a single perturbation and subsamples were not needed.

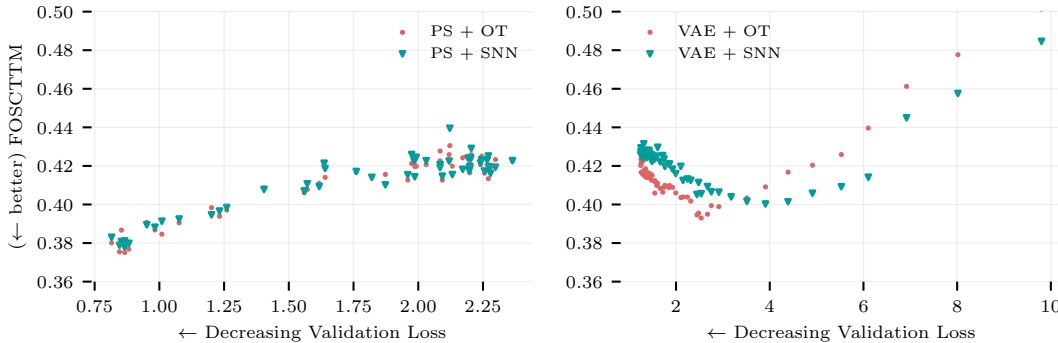

Figure 2: VAE and classifier validation metrics on the CITE-seq dataset. Notice that validation cross-entropy inversely tracks the ground truth matching metrics, and thus can be used as a proxy in practical settings where the ground truth is unknown. The same pattern does not hold for the VAE [Yang et al., 2021], which we suspect is because reconstruction is largely irrelevant for matching.

as a validation during hyperparameter tuning. Figure 2 demonstrates that the propensity score validation loss (cross-entropy) empirically satisfies this role in our CITE-seq experiments, where lower validation loss corresponds to better matching performance, as if it were computed with the ground truth. By contrast, we found that the optimal VAE, in terms of matching, had higher validation loss. This empirically supports our intuition that the reconstruction loss minimization requires the VAE to capture modality specific information, i.e., the $U^{(e)}$ variables, which hinders its matching performance.

# 7    Limitations

Our methods are limited to settings where we have some signal to play the role of an experiment label, but we believe this is where these methods are most needed. Matching is impossible in general—e.g., if you tried to match modalities that have no shared information, it would clearly fail—but our theory formally articulates both where we expect this method to succeed and its limitations. Both (A1) and (A2) are strong assumptions, but the empirical results suggest the method is fairly robust to failures.

# 8    Conclusion

This work presents a simple algorithm for aligning unpaired data from different modalities using propensity scores. The method is very general, requiring only a classifier to be trained on each modality, and demonstrates excellent matching performance, which we validate both theoretically and empirically. We also showcase the effectiveness of the matching algorithm in a downstream cross-modality prediction task, achieving *better* generalization compared to random matching, VAE-based matching, and even the ground truth matching on the evaluated dataset. This improved generalization over the ground truth may be attributed to implicitly enforcing invariance to modality-specific information; a rigorous investigation of this phenomenon would be interesting for further investigation.

# 9    Acknowledgements

We are extremely grateful for the discussions with many external collaborators and colleagues at Recursion that lead to this work. The original ideas for this work stemmed from conversations with Alex Tong with feedback from Yoshua Bengio at Mila. We received a lot of helpful feedback from all of our colleagues at Valence Labs, especially Berton Earnshaw and Ali Denton. The single cell image experiments are built on code originally written by Oren Kraus and his team.

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

# A   Relaxing (A1)

**Relaxing Assumption 1**   Consider the propensity score

$$\pi(x^{(e,t)}) = P(t|X^{(e,t)} = x^{(e,t)}) \tag{11}$$

where we do not necessarily require $U^{(e)} \perp\!\!\!\perp t \mid Z^{(t)}$, and thus we obtain

$$\pi(x^{(1,t)}) = P(t|Z^{(t)} = z^{(t)}, U^{(1)} = u^{(1)}) \neq P(t|Z^{(t)} = z^{(t)}, U^{(2)} = u^{(2)}) = \pi(x^{(2,t)}), \tag{12}$$

see the proof of Proposition 3.1 for details.

Suppose that the two observed modalities are indeed generated by a shared $\{z_i\}_{i=1}^n$, but where the indices of modality 2 are potentially permuted, and with values differing by modality specific information:

$$\{x_i^{(1,t)} = f^{(1)}(z_i, u_i^{(1)})\}_{i=1}^n, \{x_j^{(2,t)} = f^{(2)}(z_2, u_j^{(2)})\}_{j=1}^n, \tag{13}$$

where $j = \pi(i)$ denotes a permutation of the sample indices. Under (A1), we would be able to find some $j$ such that $\pi(x_i^{(1,t)}) = \pi(x_j^{(2,t)})$ for each $i$.

Matching via OT can allow us to relax (A1) in a very particular way. Consider the simple case where $t \in \{0, 1\}$, so that $\pi$ can be written in a single dimension, e.g., $P(t = 1|X^{(e,t)} = x^{(e,t)}) \in [0, 1]$. In this case, exact OT is equivalent to sorting $\pi(x_i^{(1,t)})$ and $\pi(x_j^{(2,t)})$, and matching the sorted versions 1-to-1. Under (A1), the sorted versions will be exactly equal. A relaxed version of (A1) that would still result in the correct ground truth matching is to assume that $t$ affects $U^{(1)}$ and $U^{(2)}$ differently, but that the difference is order preserving, or monotone. Denote $(\pi(x_i^{(1,t)}), \pi(x_i^{(2,t)})) = (\pi_i^{(1)}, \pi_i^{(2)})$ as the true pairing, noting that we use the same index $i$. We require the following:

$$(\pi_{i_1}^{(1)} - \pi_{i_2}^{(1)})(\pi_{i_1}^{(2)} - \pi_{i_2}^{(2)}) \geq 0, \quad \forall i_1, i_2 = 1, \ldots, n. \tag{14}$$

This says that, even if $\pi_i^{(1)} \neq \pi_i^{(2)}$, that their relative orderings will still coincide. Then, exact OT will still recover the ground truth matching. See Fig. 3 for a visual example of this type of monotonicity. For example, suppose that $t$ is a chemical perturbation of a cell, and thus $\pi_i^{(1)}, \pi_j^{(2)}$ can be seen as a measure of biological response to the perturbation, e.g., in a treated population, $\pi_{i_1} > \pi_{i_2}$ indicates samples $i_1$ had a stronger response than sample $i_2$, as perceived by the first modality indexed by $i$. Then, this monotonocity states that we should see the same $\pi_{j_1} > \pi_{j_2}$ in the other modality as well, if the samples $i_1$ and $i_2$ truly corresponded to $j_1$ and $j_2$.

## A.1   Cyclic Monotonicity

We can see the monotonicity requirement Eq. (14) as the monotonicity of the function with graph $(\pi_i^{(1)}, \pi_i^{(2)}) \in [0, 1]^2$. In higher dimensions, we require that the "graph" satisfies the following cyclic monotonicity property [Villani, 2009]:

**Definition A.1.** The collection $\{(\pi_i^{(1)}, \pi_i^{(2)})\}_{i=1}^n$ is said to be $c$-cyclically monotone for some cost function $c$, if for any $n = 1, \ldots, N$, and any subset of pairs $(\pi_1^{(1)}, \pi_1^{(2)}), \ldots, (\pi_n^{(1)}, \pi_n^{(2)})$, we have

$$\sum_{n=1}^{N} c(\pi_n^{(1)}, \pi_n^{(2)}) \leq \sum_{n=1}^{N} c(\pi_n^{(1)}, \pi_{n+1}^{(2)}). \tag{15}$$

Importantly, we define $\pi_{n+1} = \pi_1$, so that the sequence represents a cycle.

Note in our setting, the OT cost function is the Euclidean distance, $c(x, y) = \|x - y\|_2$. It is known that the OT solution must satisfy cyclic monotonicity. Thus, if the true pairing is uniquely cyclically monotone, we can recover it with OT. However, we are unaware of common violations of (A1) that would satisfy cyclic monotonicity.

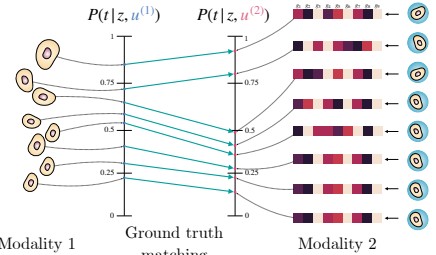 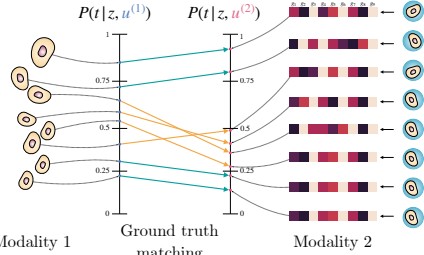

Figure 3: OT matching allows for $t$ to have different effects on the modality specific information, here $u_i^{(1)}$ and $u_i^{(2)}$, as long as they can be written as transformations that preserve the relative order within modalities. Exact OT in 1-d always matches according to the relative ordering, and thus exhibits this type of "no crossing" behaviour shown in the figure on the left. The figure on the right shows a case where we would fail to correctly match across modalities because of the crossing shown in orange.

## B  Shared Nearest Neighbours Matching

Using the propensity score distance, we can compute nearest neighbours both within and between the two modalities. We follow Cao and Gao [2022] and compute the normalized shared nearest neighbours (SNN) between each pair of observations as the entry of the matching matrix. For each pair of observations $(\pi_i^{(1)}, \pi_j^{(2)})$, we define four sets:

- $11_{ij}$: the k nearest neighbours of $\pi_i^{(1)}$ amongst $\{\pi_i^{(1)}\}_{i=1}^{n_1}$. $\pi_i^{(1)}$ is considered a neighbour of itself.
- $12_{ij}$: the k nearest neighbours of $\pi_j^{(2)}$ amongst $\{\pi_i^{(1)}\}_{i=1}^{n_1}$.
- $21_{ij}$: the k nearest neighbours of $\pi_i^{(1)}$ amongst $\{\pi_j^{(2)}\}_{j=1}^{n_2}$.
- $22_{ij}$: the k nearest neighbours of $\pi_j^{(2)}$ amongst $\{\pi_j^{(2)}\}_{j=1}^{n_2}$. $\pi_j^{(2)}$ is considered a neighbour of itself.

Intuitively, if $\pi_i^{(1)}$ and $\pi_j^{(2)}$ correspond to the same underlying propensity score, their nearest neighbours amongst observations from each modality should be the same. This is measured as a set difference between $11_{ij}$ and $12_{ij}$, and likewise for $21_{ij}$ and $22_{ij}$. Then, a modified Jaccard index is computed as follows. Define

$$J_{ij} = |11_{ij} \cap 12_{ij}| + |21_{ij} \cap 22_{ij}|, \tag{16}$$

the sum of the number of shared neighbours measured in each modality. Then, we compute the following Jaccard distance to populate the unnormalized matching matrix:

$$\tilde{M}_{ij} = \frac{J_{ij}}{4k - J_{ij}}, \tag{17}$$

where notice that $4k = |11_{ij}| + |12_{ij}| + |21_{ij}| + |22_{ij}|$, since each set contains $k$ distinct neighbours, and thus $0 \leq \tilde{M}_{ij} \leq 1$, as with the standard Jaccard index. Then, we normalize each row to produce the final matching matrix:

$$M_{ij} = \frac{\tilde{M}_{ij}}{\sum_{i=1}^{n_1} \tilde{M}_{ij}}. \tag{18}$$

Note $M_{ij}$ is always well defined because $\pi_i^{(1)}$ and $\pi_j^{(2)}$ are always considered neighbours of themselves.

**Lemma B.1.** $\tilde{M}_{ij}$ *has at least one non-zero entry in each of its rows and columns for any number of neighbours* $k \geq 1$.

*Proof.* We prove that $J_{ij} > 0$ for at least one $j$ in each $i$, which is equivalent to $\tilde{M}_{ij} > 0$. Fix an arbitrary $i$. $21_{ij}$ by definition is the same set for every $j$. By the assumption of $k \geq 1$ it is non-empty,

so there exists $\pi_{j*}^{(2)} \in 21_{ij}$. Since $\pi_{j*}^{(2)}$ is a neighbour of itself, we have $\pi_{j*}^{(2)} \in 22_{ij*}$, showing that $J_{ij*} > 0$. The same reasoning applied to $11$ and $12$ also shows that $J_{ij}$ for at least one $i$ in each $j$. $\qquad\square$

## C Proofs

### C.1 Proof of Proposition 3.1

*Proof.* Let $x^{(e)}$ denote the observed modality and $z, u^{(e)}$ be the unique corresponding latent values. By injectivity,

$$
\begin{aligned}
\pi(x^{(e)}) &= P(t|X^{(e)} = x^{(e)}) \\
&= P(t|Z = z, U^{(e)} = u^{(e)}) \\
&= P(t|Z = z) = \pi(z),
\end{aligned}
\tag{19}
$$

for $e = 1, 2$, since we assumed $U^{(e)} \perp\!\!\!\perp t \mid Z$. Since this holds pointwise, it shows that $\pi(X^{(1)}) = \pi(X^{(2)}) = \pi(Z)$ as random variables. Now, a classical result of Rubin [1974] gives that $Z \perp\!\!\!\perp t \mid \pi(Z)$, and that for any other function $b$ (a *balancing score*) such that $Z \perp\!\!\!\perp t \mid b(Z)$, we have $\pi(Z) = g(b(Z))$. The first property written in information theoretic terms yields,

$$
I(t, Z \mid \pi(Z)) = I(t, Z \mid \pi(X^{(e)})) = 0,
\tag{20}
$$

since $\pi(X^{(e)}) = \pi(Z^{(t)})$ as random variables, as required. $\qquad\square$

### C.2 Proof of Proposition 3.2

*Proof.* In what follows, we write $\pi$ to be the restriction to its domain where it is strictly positive. The $i$-th dimension of the propensity score can be written as

$$
(\pi(z))_i = p(t = i|z) = \frac{p(z|t = i)p(t = i)}{\sum_{i=0}^{T} p(z|t = i)p(t = i)},
\tag{21}
$$

which, when restricted to be strictly positive, maps to the relative interior of the $T$-dimensional probability simplex. Consider the following transformation:

$$
h(\pi(z))_i = \log\left(\frac{(\pi(z))_i}{(\pi(z))_0}\right)
\tag{22}
$$

$$
= \log(p(z|t = i)) - \log(p(z|t = 0)) + C,
\tag{23}
$$

where $C = \log(p(t = i)) - \log(p(t = 0))$ is constant in $z$, and that $h(\pi(z))_0 \equiv 0$. Ignoring the constant first dimension, we can view $h$ as an invertible map to $\mathbb{R}^T$. Under this convention, the map $h \circ \pi : \mathbb{R}^d \to \mathbb{R}^T$ is smooth (log is smooth, and the densities are smooth by assumption). Since it is smooth, it cannot be injective if $T < d$ [Spivak, 2018]. Finally, since $h$ is bijective, this implies that $\pi$ cannot be injective. $\qquad\square$

## D Experimental Details

### D.1 Evaluation Metrics

#### D.1.1 Known Ground Truth

In the synthetic image and CITE-seq datasets, a ground truth matching is known, and we can evaluate the quality of the synthetic matching directly against the truth. In these cases, the dataset sizes are necessarily balanced, so that $n = n_1 = n_2$. In each case, we evaluate the quality of our $n \times n$ matching matrix $M$, which we compute within samples with the same $t$. Our reported results are then averaged over each cluster. Note we randomize the order of the datasets before performing the matching to avoid pathologies.

**Trace Metric**  Assuming the sample indices correspond to the true matching, we can compute the average weight on correct matches, which is the normalized trace of $M$:

$$\frac{1}{n}\text{Tr}(M) = \frac{1}{n}\sum_{i=1}^{n} M_{ii}. \tag{24}$$

As a baseline, notice that a uniformly random matching that assigns $M_{ij} = 1/n$ for each cell yields $\text{Tr}(M) = 1$ and hence will obtain a metric of $1/n$. This metric however does not capture potential failure modes of matching. For example, exactly matching one sample, while adversarially matching dissimiliar samples for the remainder also yields a trace of $1/n$, which is equal to that of a random matching.

**Latent MSE**  On the image dataset, we have access to the ground truth latent values that generated the images, $\mathbf{z} = \{z_i\}_{i=1}^{n}$. We compute matched latents as $M\mathbf{z}$, the barycentric projection according to the matching matrix. Then, to evaluate the quality of the matching in terms of finding similar latents, we compute the MSE:

$$\text{MSE}(M) = \frac{1}{n}\|\mathbf{z} - M\mathbf{z}\|_2^2. \tag{25}$$

**FOSCTTM**  We do not have access to ground truth latents in the CITE-seq dataset, so use the Fraction Of Samples Closer Than the True Match (FOSCTTM) [Demetci et al., 2022, Liu et al., 2019] as an alternative matching metric. First, we use $M$ to project $\mathbf{x}^{(2)} = \{x_j\}_{j=1}^{n}$ to $\mathbf{x}^{(1)} = \{x_i\}_{i=1}^{n}$ as $\hat{\mathbf{x}}^{(1)} = M\mathbf{x}^{(2)}$. Then, we can compute a cross-modality distance as follows. For each point in $\hat{\mathbf{x}}^{(1)}$, we compute the Euclidean distance to each point in $\mathbf{x}^{(1)}$, and compute the fraction of samples in $\mathbf{x}^{(1)}$ that are closer than the true match. We also repeat this for each point in $\mathbf{x}^{(1)}$, computing the fraction of samples in $\hat{\mathbf{x}}^{(1)}$ in this case. That is, assuming again that the given indices correspond to the true matching, we compute:

$$\text{FOSCTTM}(M) =$$

$$\frac{1}{2n}\Bigg[\sum_{i=1}^{n}\left(\frac{1}{n}\sum_{j\neq i}\mathbb{1}\{d(\hat{\mathbf{x}}_i^{(1)}, \mathbf{x}_j^{(1)}) < d(\hat{\mathbf{x}}_i^{(1)}, \mathbf{x}_i^{(1)})\}\right) \tag{26}$$

$$+ \sum_{j=1}^{n}\left(\frac{1}{n}\sum_{i\neq j}\mathbb{1}\{d(\mathbf{x}_j^{(1)}, \hat{\mathbf{x}}_i^{(1)}) < d(\mathbf{x}_j^{(1)}, \hat{\mathbf{x}}_j^{(1)})\}\right)\Bigg], \tag{27}$$

where notice that this evaluates $M$ through the computation $\hat{\mathbf{x}}^{(1)} = M\mathbf{x}^{(2)}$. As a baseline, we should expect a random matching, when distances between points are randomly distributed, to have an FOSCTTM of $0.5$.

**Prediction Accuracy**  We also trained a cross-modality prediction (translation) model $f_{\theta,M}$ to predict CITE-seq protein levels from gene expression based on matched pseudosamples. Let $\mathbf{x}^{(1)} = \{x_i\}$, $\mathbf{x}^{(2)} = \{x_j\}$ denote protein and gene expression, respectively. We trained a simple 2-layer MLP minimizing either the standard MSE, using pairs $(x_i, \hat{x}_j)$, $\hat{x}_j \sim M_{i\cdot}$, or following the projected loss with unbiased estimates in Section 5. Each batch in general consists of samples from all $t$, but the $\hat{x}_j$ sampling step occurs within the perturbation. Let $\hat{\mathbf{x}}_{test}^{(1)} = \{f_{\theta,M}(x_j)\}$. We report the $R^2$ on a randomly held-out test set of ground truth pairs (again, consisting of samples from all $t$), which is defined as the following:

$$R^2(f_{\theta,M}) = \frac{MSE(\mathbf{x}_{test}^{(1)}, \hat{\mathbf{x}}_{test}^{(1)})}{MSE(\mathbf{x}_{test}^{(1)}, \bar{\mathbf{x}}_{test}^{(1)})}, \tag{28}$$

where $\bar{\mathbf{x}}_{test}^{(1)}$ is the naive mean (over all perturbations) estimator which acts as a baseline.

### D.1.2   Unknown Ground Truth

We train a cross-modality prediction model to predict gene expression from cell images based on matched pseudosamples in the same way as in CITE-seq, but only using the projected loss with unbiased estimates. Denote this model for a matching matrix $M$ by $f_{\theta,M}$.

Because we do not have access to ground truth pairs within each perturbation, we resort to distributional metrics. Let $\mathbf{x}^{(1)}_t = \{x_{i,t}\}_{i=1}^{n_{t1}}$, $\mathbf{x}^{(2)}_t = \{x_{j,t}\}_{j=1}^{n_{t2}}$ denote gene expression and cell images in a held out test set respectively in perturbation $t$. Let $\hat{\mathbf{x}}^{(1)}_t = \{f_{\theta,M}(x_{j,t})\}_{j=1}^{n_{t2}}$. We compute empirical versions of statistical divergences

$$D_t(f_{\theta,M}) := D(\mathbf{x}^{(1)}_t, \hat{\mathbf{x}}^{(1)}_t), \tag{29}$$

where $D$ is either the L2 norm of the difference in empirical mean, empirical Kullback-Leibler divergence or 1-Wasserstein distance. We report these weighted averages of $D_t$ over the perturbations $t$ according to the number of samples in the modality of prediction interest.

### D.2 Models

In this section we describe experimental details pertaining to the propensity score and VAE [Yang et al., 2021]. SCOT [Demetci et al., 2022] and scGLUE [Cao and Gao, 2022] are used according to tutorials and recommended default settings by the authors.

**Loss Functions**  The propensity score approach minimizes the standard cross-entropy loss for both modalities. The VAE includes, in addition to the standard ELBO loss (with parameter $\lambda$ on the KL term), two cross-entropy losses based on classifiers from the latent space: one, weighted by a parameter $\alpha$ to classify $t$ as in the propensity score, and another, weighted by a parameter $\beta$, that classifies which modality the latent point belongs to.

**Hyperparameters and Optimization**  We use the Adam optimizer with learning rate 0.0001 and one cycle learning rate scheduler. We follow Yang et al. [2021] and set $\alpha = 1$, $\beta = 0.1$, but found that $\lambda = 10^{-9}$ (compared to $\lambda = 10^{-7}$ in Yang et al. [2021]) resulted in better performance. We used batch size 256 in both instances and trained for either 100 epochs (image) or 250 epochs (CITE-seq).

For the VAE and classifiers of experiment 3, we use an Adam optimizer with learning 0.001 and weight decay 0.001 and max epoch of 100 (PerturbSeq) and 250 (single cell images) using batch sizes of 256 and 2048 correspondingly. We follow similar settings as Yang et al. [2021] and implement $\alpha = 1$ with $\lambda = 10^{-9}$, and since we do not have matched data, $\beta = 0$. For the cross-modal prediction models in experiment 3, we use Stochastic Gradient Descent optimizer with learning rate 0.001 and weight decay 0.001 with max epochs 250 and batch size 256. We implement early stopping with delay of 50 epochs which we then checkpoint the last model to use for downstream tasks

**Architecture**  For the synthetic image dataset, we use an 5-layer convolutional network (channels $= 32, 54, 128, 256, 512$) with batch normalization and leaky ReLU activations, with linear heads for classification (propensity score and VAE) and posterior mean and variance estimation (VAE). For the VAE, the decoder consists of convolutional transpose layers that reverse those of the encoder.

For the CITE-seq dataset, we use a 5-layer MLP with constant hidden dimension 1024, with batch normalization and ReLU activations (adapted from the fully connected VAE in Yang et al. [2021]) as both the encoder and VAE decoder. We use the same architecture for both modalities, RNA-seq (as we process the top 200 PCs) and protein.

For the PerturbSeq classifier encoder, we use a 2-layer MLP architecture. Each layer consists of a linear layer with an output feature dimension of 64, followed by Rectified Linear Unit (ReLU) activation, Batch Normalization, and dropout (p=0.1). A final layer with Leaky ReLU activation that brings dimensionality to 128 before feeding into a linear classification head with an output feature dimension of 25.

For the single-cell image encoder classifier, we use a proprietary Masked Autoencoder [Kraus et al., 2023] to generate 1024-dimensional embeddings. Subsequently, a 2-layer MLP is trained on these embeddings. Each MLP layer has a linear layer, Batch Normalization, and Leaky ReLU activation. The output feature dimensions of the linear layers are 512 and 256, respectively, and the latent dimension remains at 1024 before entering a linear classification head with an output feature dimension of 25.

**Optimal Transport**  We used POT [Flamary et al., 2021] to solve the entropic OT problem, using the log-sinkhorn solver, with regularization strength $\gamma = 0.05$.

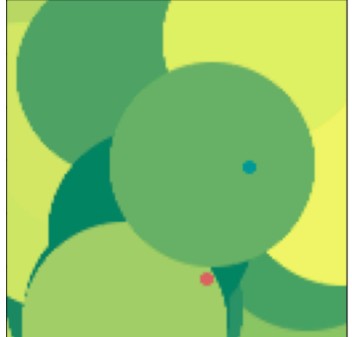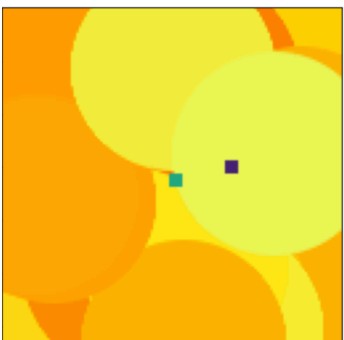

Figure 4: Example pair of synthetic images with the same underlying $z$.

## D.3 Data

**Synthetic Data**   We follow the data generating process Eq. (1) to generate coloured scenes of two simple objects (circles, or squares) in various orientations and with various backgrounds. The position of the objects are encoded in the latent variable $z$, which is perturbed by a do-intervention (setting to a fixed value) randomly sampled for each $t$. Each object has an $x$ and $y$ coordinate, leading to a 4-dimensional $z$, for which we consider 3 separate interventions each, leading to 12 different settings. The modality then corresponds to whether the objects are circular or square, and a fixed transformation of $z$, while the modality-specific noise $U$ controls background distortions. Scenes are generated using a rendering engine from PyGame as $f^{(e)}$. Example images are given in Fig. 4.

**CITE-seq Data**   We also use the CITE-seq dataset from Lance et al. [2022] as a real-world benchmark (obtained from GEO accession GSE194122). These consist of paired RNA-seq and surface level protein measurements, and their cell type annotations over 45 different cell types. We used scanpy, a standard bioinformatics package, to perform PCA dimension reduction on RNA-seq by taking the first 200 principal components. The protein measurements (134-dimensional) was processed in raw form. For more details, see Lance et al. [2022].

**PerturbSeq and Single Cell Image Data**   We collect single-cell PerturbSeq data (200 genes) and single-cell painting images in HUVEC cells with 24 gene perturbations and a control perturbation, resulting in 25 labels for matching across both modalities. The target gene perturbations are selected based on the 24 genes with the highest number of cells affected by the CRISPR guide RNAs targeting those genes. The PerturbSeq data is filtered to include the top 200 genes with the highest mean count, then normalized and log-transformed. The single-cell painting images are derived from multi-cell images, with each single-cell nucleus centered within a 32x32 pixel box. We use scVI Lopez et al. [2018] to embed the raw PerturbSeq counts into a 128-dimensional space before training the gene expression classifier. Similarly, we train our image classifier using 1024-dimensional embeddings obtained from a pre-trained Masked Autoencoder Kraus et al. [2023], He et al. [2022]. Following matching, we perform cross-modality translation from the single-cell embeddings to the transformed gene expression counts.

## D.4 Supplementary Results

| Method | In Distribution | | Out of Distribution | |
|---|---|---|---|---|
| | **Wasserstein-1 (↓)** **Med (Q1, Q3)** | **L2 Norm (↓)** **Med (Q1, Q3)** | **Wasserstein-1 (↓)** | **L2 Norm (↓)** |
| PS+OT | **4.199** **(4.173, 4.226)** | **3.280** **(3.267, 3.284)** | **5.394** | **7.219** |
| VAE+OT | 4.339 (4.314, 4.348) | 3.490 (3.486, 3.495) | 5.629 | 7.444 |
| Random | 4.499 (4.478, 4.525) | 3.826 (3.823, 3.828) | 6.239 | 7.793 |

Table 3: Wasserstein-1 and L2 norm distance values for PerturbSeq and single cell images experiments where distance is evaluated between cross-modal predictions and actual gene expression values.

