# OpenReview forum: "Propensity Score Alignment of Unpaired Multimodal Data"
_NeurIPS.cc/2024/Conference — NeurIPS 2024 poster_

### Official Review · Reviewer_qEyT · 2024-07-01

**Soundness:** 3
**Presentation:** 2
**Contribution:** 2
**Rating:** 6
**Confidence:** 4

**Summary:**

This work addresses the problem of aligning unpaired samples from multimodal data, whereby the problem is to find sample $x_i$ from modality $i$ that is best “related” to sample $x_j$ from modality $j$, when those two samples come from the observation of the same phenomenon according to two different modalities (e.g. sensors).

This is an important problem that arise in many endeavors including biology, and that has been somehow overlooked in the literature of multimodal representation learning, whereby large quantities of paired data (images and their corresponding captions) are available.
In a nutshell, the idea proposed in this work is as follows. Provided the following assumptions are valid, that is there exists a common latent space $Z$, which is entangled with a “perturbation signal” or label $t$, but that it is independent of modality specific noise $U_{i,j}$, then it is possible to estimate a common space for matching samples through the “bridge” across modalities offered by the perturbation signal $t$.

In practice, this amounts to training a classifier to predict $t$ given one of the available modality, which allows the creation of a transformed dataset endowed with a Euclidean distance. Such distance can be used to compute a matching matrix $M$, whereby entry $(i,j)$ represents the likelihood of $x_i$ being matched to $x_j$. Matching matrix $M$ can be obtained efficiently through the lenses of optimal transport matching with entropic regularization.
It should be noted that $M$ can then be used to construct pseudo-samples for any learning task that requires paired samples $(x_i, x_j)$ by replacing them with $(x_i, \hat{x}_j)$,

where the second term is obtained through the application of the matching matrix to a structural equation $f_\theta$, which is a parametric function with learnable parameters.

Several experiments on synthetic data, and a dataset whereby ground truth pairing is available, as well as experiments where no ground truth is available, indicate that the proposed method, in combination with optimal transport matching, provides an improvement over state of the art matching methods.

**Strengths:**

* This work presents a simple method to produce a matching matrix that can be used to pair multimodal data. The algorithm only requires training of two classifiers and the execution of the Sinkhorn’s algorithm to compute the optimal transport matching.

* Experiments show that the proposed method using propensity scores in combination with optimal transport matching outperform the literature in terms of MSE and other specific metrics.

* Experimental result indicate, surprisingly, that the proposed method implicitly enforces invariant to modality specific information, which endows propensity score with an improved generalization, although this specific phenomenon is only hinted at and requires additional studies

**Weaknesses:**

* The proposed method relies on strong assumptions, which are only superficially discussed. Relaxation of the conditional independence assumption A1 is discussed in appendix A, but only for the case of a small perturbation set $t \in \{ 0,1 \}$, which allows the authors conclude that under exact optimal transport, this is equivalent to operate on order preserving effects of $t$. The validity of the method also  relies on A2, which allows computing propensity score from data alone, and not the shared latent space $Z$.

* The overall perturbation method, as well as the existence of a shared latent representation $Z$ is only vaguely described. I have looked up the key reference [Yang et al, 2021], but the lack of detail is uncomfortable. It is possible to get an intuition of how the method works, but I would prefer such important technical details to be spelled out clearly.

* There is no mention of the scalability of the proposed approach. Building a matching matrix can be a daunting task for some domains whereby a large amount of unpaired training data is available. In this case, the matching matrix could be either a big square matrix, or a very skinny and tall matrix that could pose some computational challenges. It would also be important to discuss about what would be the best operating scenario for the proposed method in terms of the number of samples per modality. In very unbalanced scenarios, would the classifiers using one or the other modality really behave similarly as per Equation (5)?

**Questions:**

* Would it be possible to explain (and eventually add to the appendix) in more detail the setup from [Yang et al, 2021] which is used in this work? More precisely, it would be great to have the definition of the multimodal autoencoder, the existence of a shared latent space, and the details of the perturbations $t$ and their effects on $Z$. This would really help formalizing better the use of $t$ as a bridge between modalities.

* Can you provide a thorough discussion about the scalability of the proposed approach, the impact of imbalance in the size of the multimodal data (e.g. $n_1 >> n_2$), and the possible application to other domains with more abundant data?

* Can you elaborate more on the baselines you compare your method to? Lines 235-239 are not detailed enough to fully appreciate the differences w.r.t. the proposed method.

======= POST REBUTTAL UPDATE =======
Dear authors, thank you for the rebuttal. I have raised my score.

**Limitations:**

Yes, the authors acknowledge that their work relies on strong assumptions, but suggest that empirical results indicate their method is robust to failures. Nevertheless, I could not find the discussion on the experimental results that clearly indicates such robustness.

---

> ### Author Rebuttal · Authors · 2024-08-07
>
> **“Strong” Assumptions**
>
> The problem that we address in this paper is obviously impossible in general: for example, if I gave you data from two modalities that are completely unrelated to each other (e.g. text from the complete works of Shakespeare, and images of cells under a microscope), there would be no correct solution to the problem because there is no shared information between the two modalities. But not all multimodal matching problems are as pathological as this: if we have shared information and some way of tying them together, then matching is possible.
>
> To our knowledge, there are no prior works tackling the matching problem for which any theoretical guarantees exist on the validity of the matching itself. E.g., Ryu et al., [2024] provides theory on obtaining OT solution itself, but does not describe whether the solution corresponds to a valid matching. In this case, the pathological example above may have an OT solution, but it would clearly not be a useful one. Similarly, Yang et al. [2021] gives no theoretical guidance on the limitations of the method. Our assumptions clearly delineate where we know the problem is solvable from settings where we have no theoretical guarantees.
>
> We also note that our method empirically outperforms all of the published baselines, so we could have chosen to just present it as a state of the art method with no mention of the inherent limitations of matching in general, but we chose to be upfront about the limitations (which likely apply to all of the baselines). NeurIPS reviewer guidelines state, “Reviewers will be specifically instructed to not penalize honesty concerning limitations,” we ask you to reconsider whether the theory that is explicit about assumptions that are sufficient for the method to work is really a weakness (we think it’s a strength!).
>
> **Shared Latent Space**
>
> We will include a more formal description of Yang et al., [2021]. Roughly, they assume the following data generating process. Each modality, indexed by $i$, is generated according to $X_i = f_i(Z, N_i)$, where $Z$ is a latent variable shared across all modalities. They fit a multi-modal generative model with a VAE assuming $P_Z$ the latent distribution is known, by enforcing that all modalities have the same latent distribution. If label information $t$ is available, they add the classification term requiring $Z$ to classify $t$ (they refer to “prior knowledge” of clusters rather than perturbations, but mathematically $t$ plays the same role).
>
> We believe our data generating process (Eq. (1)) is significantly more general in describing multimodal settings. Yang et al., [2021] require a known latent distribution, while we do not even assume knowledge of the space of $Z$, nor of the functional form of $f$ (though additional structure can yield a richer theory, such as our Prop 3.2). In our biology experiments, $Z$ is just an abstract random variable representing the cells, and $U$ an abstract notion of measurement noise. For the VAE, any theoretical guarantees would require correctly specified latent spaces and decoder functional classes (both of which are also untestable). In [Yang et al., 2021], they state
>
> > The dimensionality of the latent distribution is a hyperparameter that is tuned to ensure that the autoencoders are able to reconstruct the respective data modalities well.
>
> By contrast, our theory about propensity scores is valid with only (A1) and (A2), both of which, are easy to interpret scientifically for practitioners (e.g., (A1): interventions do not modify the measurement process, (A2): the measurement contains complete information about the biology).
>
> **Scalability**
>
> Entropy regularized OT is $\mathcal(O)(n^2 / \epsilon)$ (ignoring log factors, see [Pham 2020] for details), where in our setting, n corresponds to the number of samples in each domain (bounded by the max number of sample in the unbalanced case you refer to). Given that we only match within a treatment group, this is not unreasonable in the domains that we consider: for example, a well of cells in a cell-paint assay will typically have approximately 1000 cells. Constructing a 1000 x 1000 distance matrix and solving the resulting OT problem is very doable on current hardware. Of course, there may exist settings with even larger numbers of samples which would require more scalable unbalanced OT methods, or alternatively switching to shared nearest neighbors (at the cost of some performance; see our experiments). We emphasize that our contribution is in designing the distance metric that leverages propensity scores (and explaining theoretically when and why it works), rather than any particular matching approach that operates on that distance metric, so our approach can be combined with any scalable method that operates on pairwise distances, and we simply found that entropy regularized OT was most effective empirically.
>
> Pham, K., Le, K., Ho, N., Pham, T., & Bui, H. (2020, November). On unbalanced optimal transport: An analysis of sinkhorn algorithm.
>
> **Elaborate on the baselines**
>
> Gromov-Wasserstein OT (SCOT) uses the local geometry within each modality, assumed to lie in a metric space, to build a cross-modality distance by comparing the distances between pairs within modalities. We expect this approach to fail when one of the modalities does not have a simple metric structure, e.g., for images where euclidean distance in pixel space usually poorly captures semantic similarity. scGLUE is also a VAE approach that is tailored to biological applications, by enforcing that gene expression and their associated proteins (or, in their case, ATAC-seq locations) to have similar embeddings. As such, scGLUE is not applicable outside of genomics data. Importantly, neither of these methods utilize label information. For this reason, the VAE of [Yang et al., 2021] described above, which can use label information in addition to the VAE objective, is considered our main comparison.

---

> > ### Comment · Reviewer_qEyT · 2024-08-11
> > **Thank you for your rebuttal**
> >
> > Dear authors,
> > thank you for the thorough rebuttal to my observations and questions.
> >
> > I appreciate the explanation on the assumptions made in this work, and how do they compare to the ones done in the state of the art.
> >
> > I also appreciate the discussion on the obtained results, which surpass currently available methods.
> >
> > I will raise my score to reflect these observations.

---

> > > ### Author Response · Authors · 2024-08-11
> > >
> > > Thank you so much!

---

### Official Review · Reviewer_XqRN · 2024-07-03

**Soundness:** 2
**Presentation:** 3
**Contribution:** 2
**Rating:** 5
**Confidence:** 3

**Summary:**

The paper proposes a novel matching method for unpaired multimodal (bimodal) data. It aligns unpaired samples across modalities using a propensity score. Based on additional treatment information, the propensity score is learned and used to align multimodal samples.
The method is evaluated on three datasets: one imaging dataset and one single-cell dataset (both with available matching ground truth); and one additional single-cell microscopy dataset without available ground truth.

**Strengths:**

- Interesting idea: using a propensity score that leverages (often) available labels/treatments to learn a classifier is interesting.
- Important Problem: A long line of multimodal methods describes how to leverage best-paired samples, but little work has been done on aligning unpaired samples. Besides biology, there is (probably) a lot of unpaired multimodal data.
- well written and easy to read

**Weaknesses:**

- Final purpose of the method: To me, it was not fully clear what the final goal/purpose of the method is. Is it (1) the aligning of unpaired multimodal samples? Or (2), learning from unpaired, multimodal samples while aligning the samples? If (1) is the goal, what can we conclude from the improved results? If (2) is the goal, it would be interesting to see if the improved alignment/matching results in better downstream task/learning from multimodal data compared to other works (e.g., Yang et al., 2021) or just unimodal methods without any pairing of data.
- Evaluation: It is not straightforward to understand the used metrics (maybe also related to the point above) and relate the performance metrics to the quality of the method (besides a relative performance between different alignment methods). Maybe reporting some classification-based performances for alignment (is the assigned sample from the other modality the correct one) or some downstream task performance (what task can we solve using the trained method?) would help.
- Although the method is advertised as multimodal, no experiments or proofs use more than two modalities. Maybe the term bimodal would be more accurate? Otherwise, an experiment or derivation for more than two modalities would be appreciated.

**Questions:**

- what are the labels/perturbations in the image-based dataset? I did not see the information in section 6 (but maybe I missed it. Apologies in that case)
- section 6.2: what is the reason for using the first 200 principal components for the gene expression modality (and not for the protein data as well)?
- does assumption A1 hold in the datasets used? Is there a way to check whether the assumption holds?
- there are some typos:
    - line 90: the our instead of the or our
    - line 168: form instead of from
    - line 227: against on instead of against or on

**Limitations:**

The authors address some of the limitations.

---

> ### Author Rebuttal · Authors · 2024-08-07
>
> **Final purpose of the method**
>
> We are most interested in (2) “learning from unpaired multimodal samples”. Our cross modality task (see Table 2) explicitly evaluate this and show that you do indeed see a significant improvement by learning from matched data (the difference in performance between our approach and Yang et al. shows that one gets a larger improvement from better matching).
>
> **Evaluation**
>
> We agree that the metrics can be difficult to interpret, we will include more details in the main text for clarification.
>
> > Maybe reporting some classification-based performances for alignment (is the assigned sample from the other modality the correct one)
>
> We report two metrics that measure how close the estimated matchings are to the correct one, when the ground truth is known (image and CITE-seq data). The trace metric corresponds exactly to a measure of how often the assigned sample from the other modality is the correct one ($Tr(M) = \sum_{i} M_{ii}$, where $M_{ii}$ is the weight given to the true pairing).The FOSCTTM is a standard metric in the matching literature, where a value of 0.3 indicates that, on average, the true sample is within the closest 30% to the true sample.
>
> Both of these metrics are standard (which is why we report them), but essentially evaluate your task (1), “aligning unpaired multimodal samples”. For what you called task (2) (learning from unpaired samples), what matters more is (i) how good a proxy the matched sample is for the true match, and (ii) performance on downstream tasks. On synthetic data, we can evaluate (i) by testing how close the latent state of the matched sample is to the ground truth match (MSE in table 1). This metric is a better proxy for downstream performance because if there are a number of samples that are very similar to the true match, then it is likely that you won’t get the exact match correct, but that matching to any of the close samples will still lead to good results (because they’re good proxies for the true match). As discussed above, we also report (ii) in Table 2 (see below for more detail).
>
> > or some downstream task performance (what task can we solve using the trained method?)
>
> We emphasize that the metrics above that measure may not be indicative of downstream especially when using soft matching techniques such as OT. The $R^2$ metric on the prediction task (section 5) can be interpreted as the amount of information that is shared between modality 1 and the paired modality 2. E.g., an $R^2$ of 0.233 can be roughly interpreted as the paired modality 1 explains 23.3% of the variation in modality 2, while even the ground truth modality 1 only explains 22.4% (Table 2), indicating that the matched samples are statistically indistinguishable from the true matches in terms of shared information.
>
> **Multimodal vs. bimodal**
>
> > advertised as multimodal, no experiments or proofs use more than two modalities.
>
> Using the term “multimodal” for settings with two or more modalities is standard in the literature. For example, Manzoor et al. 2024 state,
> > Multimodal systems utilize **two or more** input modalities ... which could be different from the inputs. [emphasis added]
>
> Similarly, most of multimodal methods discussed in Guo et al [2019] involve only two modalities, in some cases with no obvious way of extending them beyond just two modalities.
>
> That said, the multimodal matching could in theory be done by iteratively applying bivariate matching to a base modality, e.g., sampling a tri-modal observation $(x_i^{(1)}, x_{j}^{(2)}, x_{k}^{(3)})$ would involve sampling $x_j^{(2)}$ from $M^{(1,2)}$ and $x_k^{(3)}$ from $M^{(1,3)}$. One can also think of distance functions involving multi-pairs of propensity scores, e.g., a cost function $c(\pi_i, \pi_j, \pi_k)$, which can be aligned with [multi-marginal optimal transport](https://arxiv.org/pdf/1406.0026), but our main contribution is to introduce the common space defined by the propensity score.
>
> Manzoor, M. A., Albarri, S., Xian, Z., Meng, Z., Nakov, P., & Liang, S. (2023). Multimodality representation learning: A survey on evolution, pretraining and its applications.
>
> Guo, W., Wang, J., & Wang, S. (2019). Deep multimodal representation learning: A survey.
>
> **Questions**
>
> > what are the labels/perturbations in the image-based dataset?
>
> The perturbations are do-interventions on the latents (location) of the objects in the rendered images, it is briefly described on l253.
>
> > section 6.2: reason for using the first 200 principal component?
>
> We chose to use principal components for the GEX data due to sparsity (over ~ 20k dimensions), which is common pre-processing in bioinformatics pipelines. Note this was only used as deterministic pre-processing to train the classifiers and in theory could be avoided with a flexible enough classifier (e.g., a GEX-specific encoder). The protein data is only ~130 dimensions and the measurements are naturally more dense and continuous, and we found that a classifier trained directly on raw measurements performed adequately.
>
> >. does assumption A1 hold in the datasets used? Is there a way to check whether the assumption holds?
>
> Whether or not A1 holds, depends in part on the extent to which A2 holds: any part of Z that is not reflected in modality 1 essentially becomes part of $U^{(2)}$ and vice versa. Because CITE-Seq only measures surface proteins of the cell, there is likely part of the latent cell state that is not reflected in the proteomics assays, and some of that state is surely affected by the cell type, so we should not expect assumption A1 to hold exactly. That said, the assumptions remain useful for reasoning about the conditions under which matching is theoretically possible (see also our response to R3) and guiding data collection. E.g. our theory suggests that we will have better matching performance with an assay that measures all proteins, and not just surface proteins.

---

> > ### Comment · Reviewer_XqRN · 2024-08-07
> >
> > Dear authors,
> >
> > Thanks for your rebuttals and replies to my questions.

---

> > > ### Author Response · Authors · 2024-08-07
> > >
> > > Likewise, thank you so much for the time you put into your review! Please let us know if there's anything else you need to know, and if you have any other concerns that are preventing you from raising your score.

---

> > > > ### Comment · Reviewer_XqRN · 2024-08-09
> > > >
> > > > Dear authors,
> > > >
> > > > There is a long line of work discussing truly multimodal methods, e.g. [1,2,3], which is important as many new problems arise when trying to scale methods from 2 to $M$ modalities.
> > > >
> > > > But I acknowledge your efforts in explaining and answering my questions. I will raise my score.
> > > >
> > > >
> > > > ---
> > > >
> > > > ### References
> > > > [1] Wu and Goodman, "Multimodal Generative Models for Scalable Weakly-Supervised Learning", Neurips 2018
> > > >
> > > > [2] Shi et al., "Variational mixture-of-experts autoencoders for multi-modal deep generative models", Neurips 2019
> > > >
> > > > [3] Sutter et al., "Generalized Multimodal ELBO", ICLR 2021

---

### Official Review · Reviewer_iHqN · 2024-07-15

**Soundness:** 3
**Presentation:** 3
**Contribution:** 3
**Rating:** 7
**Confidence:** 3

**Summary:**

This paper presents a new method for pairing unpaired examples across different modalities using the labels of the examples. The method essentially trains a classifier for predicting labels for examples in each modality, then uses the classifier's logits across two modalities to calculate a similarity matrix. Empirical results show that the method is promising in aligning data from different modalities.

**Strengths:**

- The paper is written clearly.
- The problem of aligning unpaird data in multimodal learning is quite important since in many cases there's limited or no access to paired data for representation learning. This paper proposes a new method for it, paving the way to future work in this area.
- The method is relatively straightforward and easy to follow.
- Empirical results show that the method perform well compared to the baseline.

**Weaknesses:**

The method uses the information in the labels in order to match labels via classifiers. There are two drawbacks with this approach: (1) the quality of matching depends directly on the choice of the classifier and its capacity/complexity. (2) more importantly, in many multimodal representation learning settings the label signal is unavailable.

**Questions:**

Is it possible that the use of labels as the sole signal to match examples from two modalities results in example clustering based on labels? i.e. examples from modality 1 that belong to class t are matched strongly against those from modality 2 that also belong to class t?
In that case, is the method still beneficial compared to not doing any matching (i.e. just using data from individual modalities)? Having a discussion on this in the paper will be useful.

**Limitations:**

There's a short section on limitations of the method, explaining (1) the reliance of the method on labels, and (2) the strong assumptions made in the method.

---

> ### Author Rebuttal · Authors · 2024-08-07
>
> **Dependence on a classifier**
> >There are two drawbacks with this approach: (1) the quality of matching depends directly on the choice of the classifier and its capacity/complexity.
>
> It is true that the quality of the matching depends on the choice of classifier and its capacity, but because you can easily validate the classifier’s performance on a validation set, this makes it relatively easy to choose a good classifier on which to base the matching. Training a classifier is not always trivial, but it is far easier than, e.g. training a variational autoencoder with a shared latent state as in Yang et al. [2021].
> > more importantly, in many multimodal representation learning settings the label signal is unavailable.
>
> The assumption of perturbation labels is a restriction but it is extremely common in the biological settings that we study (note the concurrent work we cited, Ryu et al. [2024], which was posted just before the submission deadline, also study the same setting motivated by its biological applications). Even when experimental data is not available, it is often possible to rely on weaker class labels. E.g. in our real data experiments on CITE-Seq data, we used cell type as a label.
>
> **Question about labels**
> > examples from modality 1 that belong to class t are matched strongly against those from modality 2 that also belong to class t”
>
> In all of our experiments, we perform matching within a known class. So all the examples from class $t$ in modality 1 are restricted to only match to examples from modality 2 in class $t$ (and vice versa). This is why random matching still gives fairly decent results on cross modality prediction tasks (Table 2); we believe that this is roughly equivalent to the “not doing any matching (i.e. just using data from individual modalities)” baseline you refer to (but please let us know if you have something else in mind so that we can update the camera ready). The additional benefit over random comes from matching within a class. Note that matching results in a significant improvement in $R^2$ and the KL metrics.

---

### Decision · Program_Chairs · 2024-09-25

**Decision:**

Accept (poster)

**Comment:**

This paper introduces a novel method for aligning unpaired multimodal data using label-based classifiers. The approach involves training classifiers to predict labels for each modality and using the resulting logits to calculate a similarity matrix between modalities. The method leverages a common latent space, influenced by a perturbation signal, to align samples across modalities. This alignment is further refined using optimal transport matching with entropic regularization. Empirical results on multiple datasets, including imaging and single-cell data, demonstrate that this method effectively improves sample alignment compared to existing techniques.

All reviewers agree that this work is interesting and important and the paper is well written and easy to follow. Experiments show that the proposed method outperforms the literature in terms of MSE and other specific metrics. I think this is an interesting paper that should be accepted for the conference.